# Active case-finding for TB in India: Assessment of scale and quality benchmarks, time taken and use of rapid molecular diagnostic tests

Hemant Deepak Shewade[1‡]*, S Kiran Pradeep[1‡], Prabhadevi Ravichandran[1], G Kiruthika[1], Amar N. Shah[2], Bhavin Vadera[2], R Sabarinathan[1], Venkatesh Roddawar[3], Sanjay K. Mattoo[4], Swati Iyer[5], Dheeraj Tumu[5], Aniket Chowdhury[5], Shanmugasundaram Devika[1], Joshua Chadwick[1], Rakesh R. Vaidya[6], Prasanta Kumar Hota[7], Asha Frederick[8], Pankaj Singh[9], Bayarilin Shanpru[10], Gunjan Khunger[11], Subrata Kumar Panda[5], Mohamed Arif Baig[5], K V Suma[5], Mahendran Suleka[1], Amit Kumar Digal[1], Debobrata Banerjee[1], Machupalli Lakshmi Prasanna[1], Divya Yashpal Waghela[1], A Krishnaraj[1], Pratibha Kashyap[1], Jayeshbhai Sendhabhai Parmar[1], Sushmita Das[1], Ajit Kumar[1], Ajay Kumar[1], Sunny Yadav[1], Shradha Chetri[1], Subhashree Suchismita Mahapatra[1], Amit Kumar[1], Monu Pathak[1], Sachin Singh[1], Shams Tabrez[1], Piyush Mehra[1], Bipra Bishnu[5], G Mahesh[5], Biswabihari Mohanty[12], A Rajesham[13], Bal Krishna Mishra[14], Dinesh N. Barot[15], Umesh Chandra Tripathi[5], Khalid Umer Khayyam[16], Kiran Rade[5‡], Raghuram Rao[4‡], Manoj V. Murhekar[1‡]

1 ICMR-National Institute of Epidemiology (ICMR-NIE), Chennai, India, 2 USAID India, New Delhi, India, 3 John Snow India Private Limited, New Delhi, India, 4 Central TB Division, Ministry of Health and Family Welfare, New Delhi, India, 5 Office of the World Health Organization (WHO) Representative to India, WHO Country Office, New Delhi, India, 6 State TB Cell, Government of Gujarat, Ahmedabad, India, 7 State TB Cell, Government of Odisha, Bhubaneshwar, India, 8 State TB Cell, Government of Tamil Nadu, Chennai, India, 9 State TB Cell, Government of Uttarakhand, Dehradun, India, 10 East Khasi Hills District TB Cell, Government of Meghalaya, Shillong, India, 11 Ganganagar District TB Cell, Government of Rajasthan, Ganganagar, India, 12 Anugul District TB Cell, Government of Odisha, Anugul, India, 13 State TB Cell, Government of Telangana, Hyderabad, India, 14 State TB Cell, Government of Bihar, Patna, India, 15 Kheda District TB Cell, Government of Gujarat, Nadiad, India, 16 National Institute of Tuberculosis and Respiratory Diseases, Government of India, New Delhi, India

‡ HDS and SKP joint primary authors; KR, RR and MVM joint senior authors
* hemantjipmer@gmail.com

## Abstract

Since 2017, tuberculosis active case-finding (TB ACF) has been implemented within the routine framework of India's national TB elimination program. Symptom screen of high-risk population followed by confirmation of TB among symptom-screen positive is the algorithm. ACF scale and quality assessments hitherto were predominantly local or based on aggregate program data with limited details on all the scale and quality indicators, the time taken and the extent of use of rapid molecular diagnostic tests in ACF care cascade. In this cohort study from high-risk populations in 30 randomly sampled districts (nine states), we assessed one ACF cycle (intention to screen the high-risk population once) during January-September 2023 using prospectively collected individual level data. 581 633 high-risk individuals were screened by the program utilizing existing workforce and resources. The two scale indicators

**Data availability statement:** All the data are fully available without restriction. Figshare: The link contains a collection of the dataset (EpiData REC file), the codebook/data dictionary for dataset (MS Excel) and the programme file for analysis of dataset (EpiData PGM file) pertaining to this study. https://doi.org/10.6084/m9.figshare.c.7048385.v1 Data are available under the terms of the Creative Commons Attribution 4.0 International License (CC BY 4.0).

**Funding:** This study was funded by USAID through John Snow International under the TB Implementation Framework Agreement (TB commitment grants 0011-0549-1024 and 0011-0549-1025 to MVM). All authors had full access to all the data and had final responsibility for the decision to submit for publication.

**Competing interests:** The authors have declared that no competing interests exist.

(target, observed) were: percentage of i) population mapped as high-risk (≥11%, 18.3%) and ii) mapped population screened (≥90%, 7.4%). The four quality indicators (target, observed) were: percentage of i) screened identified as presumptive TB (≥5%, 2%) ii) presumptive TB tested (≥95%, 66.3%) iii) tested diagnosed as TB (≥5%, 1.6%) and iv) diagnosed put on treatment (≥95%, 100%). The number needed to screen (target ≤1538 considering the algorithm) to detect one person with TB was 4971. The same was observed across most of the high-risk groups, with few exceptions. The extent of using rapid molecular diagnostic tests was 26.4% and the median time taken from screening to sputum collection and testing was one day. To conclude, ACF scale and quality in 2023, assessed using prospectively collected individual level data, were grossly below the benchmark and lower than previously reported (2021) based on retrospective aggregate program data. Effective planning, resource allocation including use of rapid molecular diagnostic tests and individual data recording among those screened to facilitate implementation monitoring are recommended.

## Introduction

Globally, significant progress has been made in reducing the missing people with tuberculosis (TB) from 4.3 million in 2015 to 3.1 million in 2022 [1,2]. To detect the missing people with TB and put them on treatment, other case-finding strategies like enhanced and active case-finding (ACF) are employed to complement the ongoing passive case-finding (PCF) [2,3]. ACF is systematic screening for TB disease that is implemented outside of health facilities. This health system initiated case-finding includes household contact investigation and community-based systematic screening of high-risk populations [4]. World Health Organization recommends conducting ACF in high-risk populations (not in general population) that have at least 0.5% prevalence of undetected TB [3].

Many systematic reviews have synthesized information regarding effectiveness of ACF in high-risk populations, overall and in migrants, household contacts and prisons [5–8]. Even in low TB incidence countries, targeting high-risk populations has been found to be a cost-effective strategy [9]. The effectiveness of ACF on TB transmission depends on the frequency of ACF cycles per year (number of times high-risk population are screened in a year) and yield of an ACF cycle (extent of estimated undetected TB in the high-risk population that is detected because of ACF). Yield of an ACF cycle depends on the choice of high-risk populations, the scale (proportion of high-risk population in a geographic area screened) and quality (NNS - number needed to screen) of an ACF cycle, and use of sensitive screening algorithms [10,11]. Most studies assessing ACF care cascade focus on quality (NNS) and number needed to test (test-positivity). There are limited studies that describe frequency, scale and yield.

Globally, India has the highest TB burden with 0.5 million missing TB in 2022 [2,12]. A systematic review (2022) revealed that ACF implementation is highly variable in India and current literature is sparse for many important risk groups [13].

Additionally, studies focussing on ACF care cascade from India have three limitations; either they are local with limited scope for extrapolating the findings nationally [14–21], or are based in project settings [20–23], or based on aggregate retrospective secondary data [24,25]. We have limited data on all the scale and quality indicators, the time taken, and the extent of use of rapid molecular diagnostic tests in the care cascade [14–25].

In 2022, India's National TB Elimination Program (NTEP) commissioned a national level TB ACF evaluation project (August 2022 to February 2024) to guide evidence-based strategic planning. Under the first phase of this project, we assessed the frequency, scale and quality of ACF at district, state and national level for the year 2021 using retrospective secondary aggregate ACF data from *Ni-kshay* (a case-based, web-based electronic TB information management system under NTEP) [24]. Data on high-risk populations, ACF-detected presumptive TB and ACF-detected TB initiated on treatment were not available to calculate all the scale and quality indicators (see Fig 1 for all program-recommended indicators). Hence, three revised indicators based on the available data were used (see S1 Table) [24]. The findings revealed that most of the states implemented one ACF cycle, with suboptimal scale and quality (S1 Table) [24]. Though ≈9% (revised target 10% for scale) of district population was reported to be screened, we were not sure whether these were high-risk populations [24].

A nationally representative prospective study involving individual-level data from ACF cycle with reliable estimates of all scale and quality indicators (see Fig 1) was required. Recording individual data of those screened allows random checks of the aggregate numbers reported in ACF care cascade (facilitates implementation monitoring). This would add to our existing knowledge and suggest modifications required in the national ACF guidance [26,27]. Hence, under the third phase of the NTEP commissioned national level TB ACF evaluation project, we estimated at national level (2023), i) all the ACF scale and quality indicators, overall and stratified by key high-risk populations ii) the time taken in the care cascade and iii) the extent of use of rapid molecular diagnostic tests among ACF-detected people with presumptive TB.

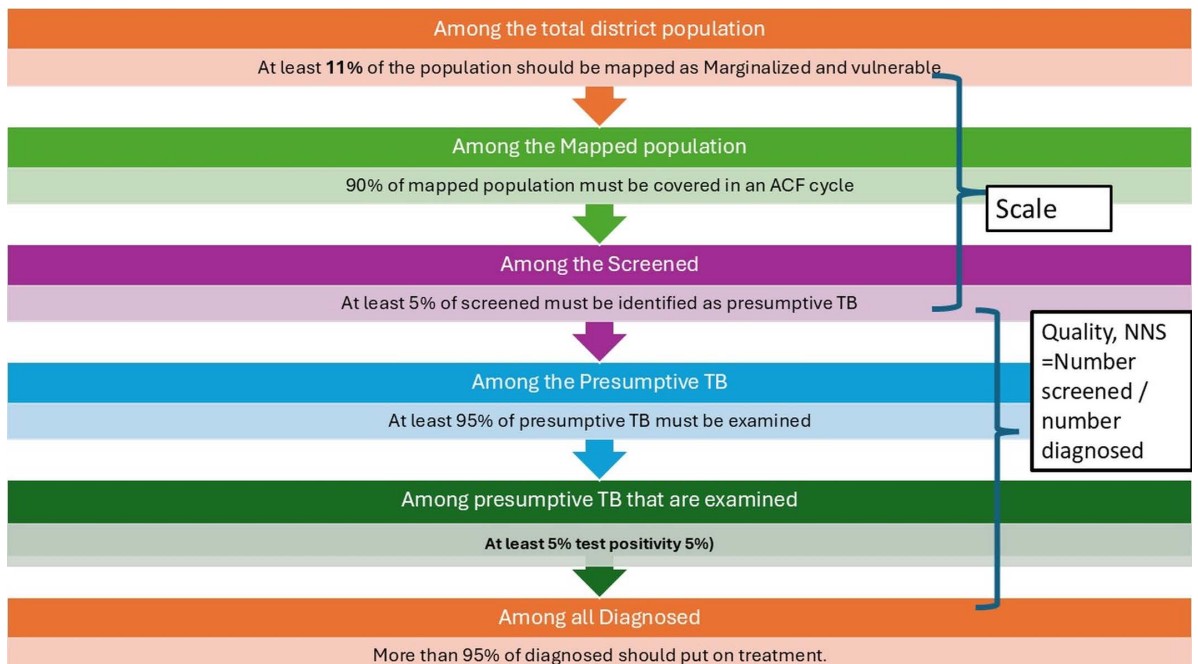

**Fig 1. Scale and quality indicators for an ACF cycle for TB, India, as per the national ACF guidance 2019 [26].** Abbreviations: ACF, active case-finding; TB, tuberculosis. Reprinted with permission from Shewade HD et al [24] under a CC BY license.

## Methods

### Study design

This was a cohort study using individual level secondary data.

### Study setting

**General setting.** India has 36 state-level TB cells and 768 NTEP districts. The central TB division provides guidance and leadership to state and district TB cells [12]. In public sector, the lowest level of peripheral health institution (PHI) facilitating TB diagnosis is the primary health centre that has at least one medical doctor and laboratory technician for sputum examination. Each sub-district level administrative unit has a senior TB treatment supervisor who captures patient notification, management and treatment outcome in *Ni-kshay* [28]. People with TB receive daily ambulatory directly observed treatment by a health care provider, community volunteer or a family member.

**Specific setting.** Since 2017, India is implementing ACF in high-risk populations of all the districts [27]. All high-risk populations are to be identified (mapped) at the start of the year and added to *Ni-kshay* [27]. High-risk populations are broadly classified into clinically, socio-economically and geographically high risk (see Table 1). Against each mapped population, aggregate numbers linked to each ACF activity date are captured: number screened, number tested and number diagnosed (number of ACF-detected presumptive TB is not captured). ACF activity days are the days identified by the district in the year when ACF will be done, irrespective of the mode (campaign mode, in multiple rounds or on fixed days of a month) or scale.

The algorithm used for ACF under NTEP involves symptom-screen (among high-risk populations) followed by sputum microscopy or rapid molecular test (Xpert MTB/Rif or TrueNat) among symptom-screen positive, also called as ACF-detected presumptive (pulmonary) TB. Symptom-screen positive is defined as the presence of cough or fever of two or more weeks or unexplained weight loss or chest pain or haemoptysis. Among clinically vulnerable, cough or fever of any duration is considered [26,27].

The national ACF guidance provides benchmarks for ACF scale and quality (Fig 1) but not for time taken and extent of use of rapid molecular diagnostic tests in the care cascade [26,27]. The 2017 ACF guidance recommended the use of

**Table 1. Specific high-risk populations for active case-finding for TB classified[a] into three broad categories.**

| Clinical | Socioeconomic | Geographical (related to health access) |
|---|---|---|
| • Contacts of people with TB<br>• Undernutrition<br>• HIV<br>• High risk groups for HIV identified by the program<br>• Noncommunicable diseases [Diabetes, hypertension, other cardio-vascular diseases, chronic lung/liver/renal disease and their risk factors (alcohol, smoking, drug addiction, others)]<br>• People who have recovered from COVID19<br>• Immunocompromised (cancer), on palliative care<br>• Occupational risk with TB (construction site workers, weaving/glass industry workers, miners, cotton mill workers, stone crushers, beedi rolling, poultry workers, tea garden workers, health care workers)<br>• Elderly | • Prisons<br>• Old age homes, night shelters, home for destitute, orphanages, refugee camps, asylums<br>• Street children/ homeless<br>• Unorganized labour<br>• Migrant population<br>• Slums<br>• Areas with high level of poverty/ people living below the poverty line<br>• Populations known to drink raw milk/ uncooked meat<br>• Indigenous/ tribal population<br>• Sanitary workers | • Hard to reach areas<br>• Villages seeking care from traditional healers<br>• Areas with clustering of TB<br>• Areas with low TB notification (than expected) |

Abbreviations: TB, tuberculosis; HIV, human immunodeficiency virus; COVID19, corona virus disease 2019.

[a]these specific high-risk groups may also be classified into facility-based and population-based, facility-based included orphanages, prison, refugee camps, homes for the destitute, tribal school homes, old age homes, night shelters, construction site workers, cotton mill workers, mine workers, stone crushers, tea garden workers, weaving, glass workers, industrial workers, poultry workers, beedi workers.

a paper-based ACF tool in the field to collect individual level information of those screened, based on which aggregate numbers are to be generated and filled in *Ni-kshay* [27]. The ACF tool includes population level information regarding the address and the specific type of high-risk population and individual level information like name, age, gender, mobile number, symptom details, presumptive pulmonary TB (yes/no) and details of testing and TB diagnosis if found to be presumptive TB. The 2019 ACF guidance did not mention the use of this paper-based ACF tool in the field [26]. Most districts and states are implementing one ACF cycle per year with limited use of the paper-based ACF tool in the field [24,29].

## Study population

In 30 randomly sampled NTEP districts (nine states) across India (see S1 Fig), we included all persons belonging to high-risk populations (not general population) who were screened between January and September 2023 and whose information were collected in a paper-based ACF tool used in the field (mobile-based in three districts belonging to one state). Aggregate data directly entered in *Ni-kshay* without the use of ACF tool was not verifiable and therefore, not considered.

We selected nine states by stratified random sampling based on the routinely available composite TB score measuring NTEP performance and TB burden for every state (>80 score classified as high, 60–80 as medium and <60 as low performance). Three states were selected from each stratum by simple random sampling. From the nine states, 30 NTEP districts were selected based on probability proportionate to size sampling (based on TB notification). Three NTEP districts each were selected from six states (Tamil Nadu, Telangana, Rajasthan, Bihar, Odisha, Meghalaya), except Gujarat, Uttarakhand and Delhi, from which four districts were selected (see S1 Fig).

## Variables, source of data and data collection

During the national/ state/ district level orientation and re-orientation meetings and in writing, it was clarified by the investigator team that individual level data available from the ACF tool used in field (see S1 File) will be considered as ACF data. The ACF tool used was similar to the one recommended in the 2017 ACF guidance [27], with the addition of two questions: whether chest radiograph was used for screening (yes/no) and whether rapid molecular test used for TB diagnosis (yes/no). Use of the ACF tool provided an opportunity for cross-checking whether ACF happened in mapped high-risk populations, verify a certain proportion of those in the care cascade and provide quality control-linked incentives [29]. This provided a certain sense of reliability to the aggregate numbers generated for the ACF care cascade.

ACF was implemented in routine settings by the health system. The district NTEP team provided overall guidance in planning, training, implementation (including mapping), and monitoring. One project research assistant per district facilitated the use of ACF tool. Within a district at the beginning of the year, in each sub-district level administrative unit (block), high risk populations were mapped (population ID, address, nearest landmark, high-risk population type and high-risk population numbers). Then this information was aggregated at district level to generate total high-risk population: broad and specific high-risk populations type (Table 1). For each ACF activity day, the project research assistant captured information on whether ACF tool was used in the field and compiled the ACF tool at district level. If there was any missing field or data inconsistency, they got it corrected. Beyond this, they did not interfere with the implementation and data quality control mechanism put in place by the district NTEP team.

## Data management and analysis

Data collection and entry were monitored centrally by the project management team from ICMR-NIE. In 27 districts, the project research assistants single-entered (with adequate data entry checks) the paper-based ACF tool data (except name and contact number)into a web application with locally stored server at ICMR-National Institute of Epidemiology (ICMR-NIE), Chennai. Three districts belonging to one state directly used an existing mobile-based version of the ACF tool in the field. Here name and contact number were not extracted for analysis. Descriptive and analytical statistics were used as appropriate using EpiData Analysis software v 2.2.2.183 (EpiData Association, Odense, Denmark).

As we captured ACF data from January to September 2023 and the districts aimed to implement one ACF cycle in 2023, 75% of the mapped high-risk population (aggregate number) was considered as intended for screening (denominator) during the study period.

The two ACF scale indicators (target) were: i) percentage of population mapped as high-risk (≥11%) and ii) percentage of mapped population screened in the ACF cycle (≥90%). The four ACF quality indicators (target) were: i) percentage of screened identified as presumptive TB (≥5%) ii) percentage of presumptive TB tested (≥95%) iii) percentage of tested diagnosed as TB (test positivity ≥5%) and iv) percentage of diagnosed put on treatment (≥95%). These were (see Fig 1) in line with national ACF guidance [26]. NNS (number screened divided by number diagnosed) cut-off (≤1538) for the algorithm summarized ACF quality indicators into one indicator [24]. We calculated the six indicators and NNS, overall and stratified by high-risk populations. It was not possible to provide the two ACF scale indicator estimates by broad high-risk population types. The reason being there were many instances where a mapped population could be classified into multiple specific high-risk population types and these specific high-risk population types could be from more than one broad high-risk population types (see Table 1).

A meaningful difference of at least 10% for proportion of presumptive tested and extent of use of rapid molecular diagnostic test, and at least 2% for other proportions was considered significant if it also achieved statistical significance (p < 0.05 using chi squared test).

### Ethics

This study was part of the third phase of India central TB division commissioned national level TB ACF evaluation project. The evaluation project was approved by ICMR-NIE's ethics committee (NIE/IHEC/202201–10 dated 09 Feb 2022, renewal on 25 Jan 2023). As this study involved use of routinely collected secondary data (anonymous), waiver of written informed consent was sought and approved by the ethics committee.

### Results

The total population of 30 districts was 57 211 032. A total of 10 449 555 (18.3%) were mapped as high-risk, 75% of which (n = 7 837 170) were considered as denominator for further analysis.

There were a total of 1467 ACF activity days. ACF tool was used in the field in 1031 (70.3%) days. During the 1031 ACF activity days considered in the analysis, of the 7 837 170 high-risk population, 581 633 (7.4%, 95% CI: 7.42. 7.44) were screened (an average of 565 people screened per district per ACF activity day). Of these, 11 357 (1.95%, 95% CI: 1.92, 1.99) were identified as presumptive TB. Among those with presumptive TB, the prevalence estimates of symptoms [duration of symptoms] were as follows: cough in 70.3% [median 9 days (interquartile range: 1,15)], fever in 28.7% [median 5 days (interquartile range: 2,7)], weight loss in 8%, haemoptysis in 2.7% and chest pain in 9.2%. Of 11 357, sputum specimen was collected for 8541 (75.2%) and 7527 (66.3%, 95% CI: 65.4, 67.2) were tested, of which 117 (1.6%, 95% CI: 1.3, 1.9) were diagnosed with TB. All the ACF-detected TB were initiated on treatment. The NNS was 4971. The ACF scale and quality indicators are depicted in Fig 2 and Table 2.

Half of those screened were male, 11% were elderly and 60% resided in rural areas (see Table 2). The most common type of high-risk population (by broad categories) screened was geographically high-risk (41%). Ten percent of high-risk individuals were screened at facilities (see Table 2). Higher percentage of presumptive TB among screened was observed in elderly (5.2% versus 1.6% in ≤60 years) and clinically high-risk (4.1% versus 1.3% in geographically high-risk). The percentage of testing among presumptive TB was low across all sub-groups and ranged from 64% to 71%. Higher test positivity was observed in socioeconomically high-risk (2.9% versus 0.6% in geographically high-risk). NNS was > 1538 for all subgroups. It was as high as 18 331 for geographically high-risk (see Table 2).

The distribution of specific high-risk populations that were screened is as follows (those with 2.5% or higher contribution have been mentioned): areas with high notifications (34.1%), slums (20.6%), occupational risk (9.5%), unorganized labour

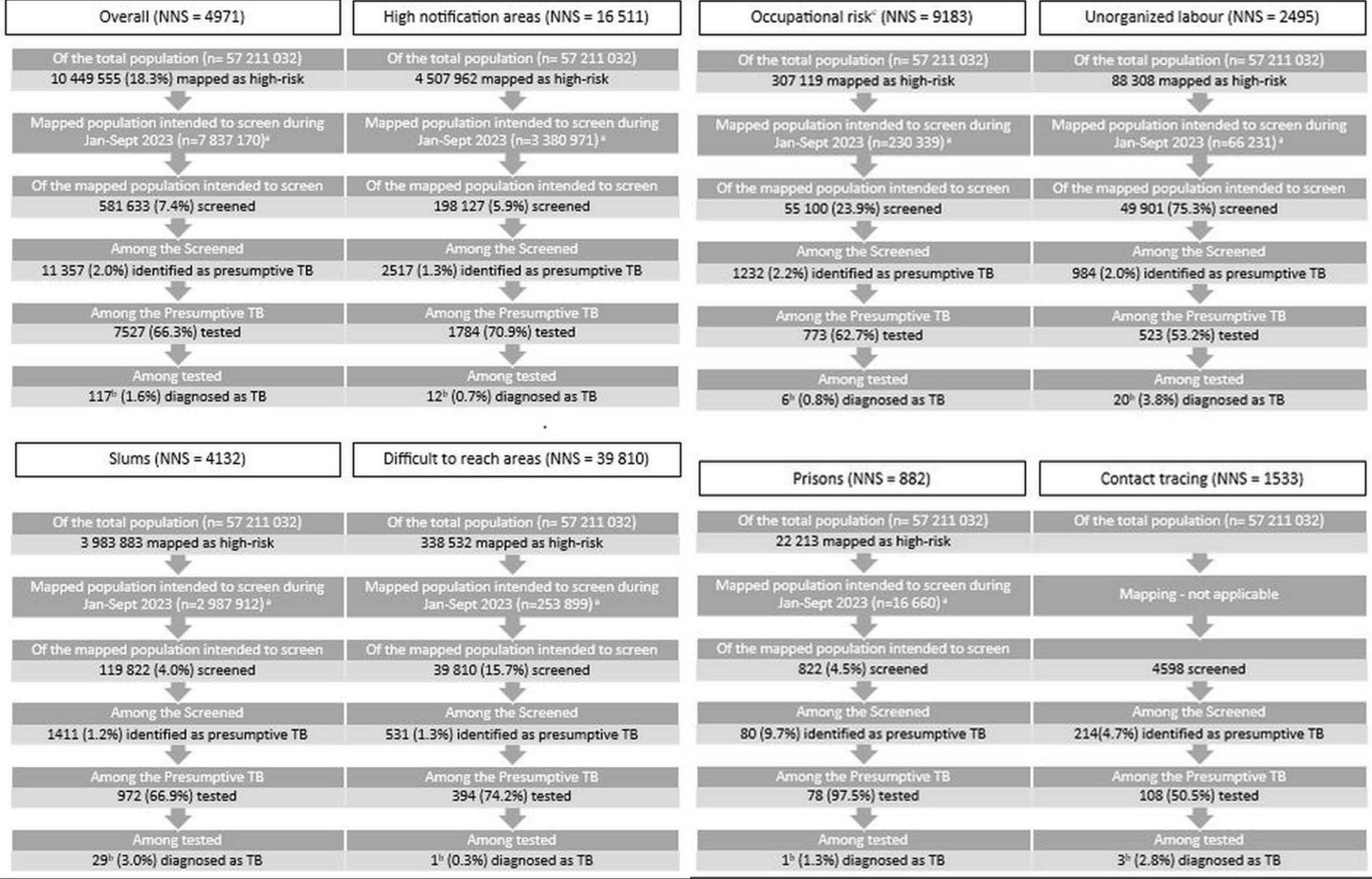

**Fig 2. Scale and quality indicators in the TB ACF cycle conducted from January to September 2023 in 30 randomly sampled NTEP districts of India, overall and stratified by specific high-risk groups** [a]**.** Abbreviations: ACF, active case-finding; TB, tuberculosis; NNS, number needed to screen (number screened divided by number diagnosed). [a]75% of the mapped high-risk population was used as denominator as we captured ACF data for three-quarters of the year (one year = one ACF cycle). Among screened, specific high-risk groups with 2.5% or higher contribution have been depicted. Data of prisons and contact screening has been included due to their importance; [b]all the ACF-detected TB were initiated on treatment; [c]occupation included construction site workers, cotton mill workers, mine workers, stone crushers, tea garden workers, weaving glass industrial workers, poultry workers, beedi workers.

(8.6%), difficult to reach areas (6.9%), tribal areas (2.9%) and noncommunicable disease and/or its risk factors and/or COVID19 recovered (2.7%). The percentage screened among the mapped population was the highest for unorganized labour (75.3%) with an NNS of 2495. The NNS was below or around the cut-off in prisons (NNS 882), contact tracing (NNS 1583) and noncommunicable disease and/or its risk factors and/or COVID19 recovered (NNS 1288) (Fig 2).

Of 7527 presumptive tested, 1989 (26.4%, 95% CI: 25.4, 27.4) underwent rapid molecular diagnostic test. High-risk populations in rural areas (18.3% versus 42.9% in urban), population-based high-risk groups (22.7% versus 43.2% for facility-based), and geographically high-risk (5.4% versus 41.1% for socioeconomically) had a significantly lower use (Table 3). The median time taken in the various steps of the care cascade ranged from zero to one day (S2 Table).

Of the high-risk population screened, a total of 9321 (1.6%) underwent upfront chest radiograph as a screening test and most of these people (n = 8854, 95%) were from two districts of a state. Of these 9321, a total of 1087 (11.7%) had an abnormality in chest radiograph, of whom 679 (62.5%) did not have any symptoms.

**Table 2. Quality indicators in the TB ACF cycle conducted from January to September 2023 in 30 randomly sampled NTEP districts of India, overall and stratified by gender, age, residence, and type of high-risk groups (broad categories).**

| Category | | Screened | | Presumptive TB | | Tested | | Diagnosed[a] | | NNS |
|---|---|---|---|---|---|---|---|---|---|---|
| | | n | (%)[col] | n | (%)[row] | n | (%) [row] | n | (%) [row] | |
| **Total** | | **581 633** | **(100.0)** | **11 357** | **(2.0)** | **7527** | **(66.3)** | **117**[a] | **(1.6)** | **4971** |
| Gender | Male | 293260 | (50.4) | 5798 | (2.0) | 3961 | (68.3) | 90 | (2.3) | 3258 |
| | Female | 284612 | (48.9) | 5526 | (1.9) | 3549 | (64.4) | 27 | (0.8) | 9208 |
| | Other | 48 | (<0.1) | 1 | (2.1) | 1 | (100.0) | 0 | (0.0) | – |
| | Missing | 3713 | (0.6) | 32 | (0.9) | 16 | (50.0) | 0 | (0.0) | – |
| Elderly | Yes (>60y) | 63645 | (10.9) | 3311 | (5.2)[b] | 2258 | (68.2) | 32 | (1.4) | 1989 |
| | No | 512 127 | (88.0) | 8007 | (1.6)[ref] | 5239 | (65.4) | 85 | (1.6) | 6025 |
| | Missing | 5861 | (1.0) | 30 | (0.5) | 30 | (100.0) | 0 | (0.0) | – |
| Residence | Rural | 349300 | (60.1) | 7590 | (2.2) | 4994 | (65.8) | 57 | (1.1) | 6128 |
| | Urban | 232300 | (39.9) | 3766 | (1.6) | 2532 | (67.2) | 60 | (2.4) | 3872 |
| | Missing | 33 | (<0.1) | 1 | (3.0) | 1 | (100.0) | 0 | (0.0) | – |
| Type of high-risk population | | | | | | | | | | |
| | Socioeconomically | 188 961 | (32.5) | 2893 | (1.5) | 1910 | (66.0) | 56 | (2.9)[b] | 3374 |
| | Clinically | 88 804 | (15.3) | 3621 | (4.1)[b] | 2504 | (69.2) | 26 | (1.0) | 3416 |
| | Geographically | 238 297 | (41.0) | 3049 | (1.3)[ref] | 2178 | (71.4) | 13 | (0.6)[ref] | 18331 |
| | Not classifiable[c] | 63 011 | (10.8) | 1794 | (2.8) | 935 | (52.1) | 22 | (2.4) | 2864 |
| Facility-based high-risk group | | | | | | | | | | |
| | Yes | 55 743 | (9.6) | 1399 | (2.5) | 915 | (65.4) | 8 | (0.9) | 6968 |
| | No | 501 619 | (86.2) | 8832 | (1.8) | 5976 | (67.7) | 93 | (1.6) | 5394 |
| | Not classifiable | 24 271 | (4.2) | 1126 | (4.6) | 636 | (56.5) | 16 | (2.5) | 1517 |

Abbreviations: TB, tuberculosis; ACF, active case-finding; NNS, number needed to screen.

[a] all the ACF-detected TB were initiated on treatment.

[b] public health wise significant difference (at least 10% for proportion of presumptive tested, at least 2% for other proportions) which was statistically significant (p < 0.05 using chi squared test) was considered.

[c] there were many instances where a mapped population could be classified into multiple specific high-risk population types and these specific high-risk population types could be from more than one broad high-risk population types.

[col] column percentage; [row] row percentage; [ref] Reference.

## Discussion

### Summary of findings

This study on TB ACF scale and quality (2023) from India is possibly the first ever nationally representative study using individual level data. During the ACF cycle, less than 10% of high-risk population was screened with an NNS of ≈5000. Time taken in the care cascade was minimal and rapid molecular diagnostic tests were used in less than one-third of those tested. Of the six scale and quality indicators, but for the percentage of population mapped as high-risk and percentage of ACF-detected TB initiated on treatment, none of the other four indicators were above the benchmark.

### Comparison with previous evidence

To begin with, let us compare our findings with 2021 [24]. First, in 2021, based on secondary aggregate *Ni-kshay* data, nearly nine percent (cut-off 10%) of the population was reported to be screened indicating acceptable scale (S1 Table) [24]. However, the limitation here was that we were not sure whether those screened were from high-risk population [24]. Our estimates from this study, which are more reliable, suggest that only 7% of mapped high-risk population were

**Table 3. Use of rapid molecular diagnostic test among ACF-detected presumptive TB who were tested in 30 randomly sampled NTEP districts of India, January to September 2023, overall and stratified by gender, age, residence and type of high-risk groups (broad categories).**

| Category | | Tested | Rapid molecular diagnostic test | |
|---|---|---|---|---|
| | | N | n | (%)row |
| **Total** | | **7527** | **1989** | **(26.4)** |
| Gender | Male | 3961 | 1150 | (29.0) |
| | Female | 3549 | 835 | (23.5) |
| | Other | 1 | 1 | (100.0) |
| | Missing | 16 | 3 | (18.8) |
| Elderly | Yes (>60y) | 2258 | 657 | (29.1) |
| | No | 5239 | 1329 | (25.4) |
| | Missing | 30 | 3 | (10.0) |
| Residence | Rural | 4994 | 902 | (18.1) a |
| | Urban | 2532 | 1087 | (42.9)ref |
| | Missing | 1 | 0 | (0.0) |
| Type of high-risk population | | | | |
| | Socioeconomically | 1910 | 784 | (41.0)ref |
| | Clinically | 2504 | 783 | (31.3)a |
| | Geographically | 2178 | 114 | (5.2) a |
| | Not classifiableb | 935 | 308 | (32.9) |
| Facility-based high-risk group | | | | |
| | Yes | 915 | 395 | (43.2)ref |
| | No | 5976 | 1340 | (22.4)a |
| | Not classifiable | 636 | 254 | (39.9) |

Abbreviations: TB, tuberculosis; ACF, active case-finding; NNS, number needed to screen.

a difference in proportion of at least 10% and statistically significant (p < 0.05) using chi-squared test.

b there were many instances where a mapped population could be classified into multiple specific high-risk population types and these specific high-risk population types could be from more than one broad high-risk population types.

row row percentage; refReference.

screened (cut-off 90%). Even if we assume that the ACF tool was used in all ACF days and the mapped high-risk population was reduced to 11% (instead of 18.3% observed), the estimated scale of ACF will increase only to 17.5%, still way lower than the target of 90%.

Second, in 2021, 1% of screened were tested against the target of 4.75% (S1 Table) [24]. Our findings in 2023 that provide information on all ACF quality indicators suggest that it is due to a combination of sub-optimal screening in the field (low percentage of presumptive among those screened) and losses between presumption, sputum collection and testing. Not capturing aggregate number of presumptive among screened in *Ni-kshay* could be the reason for missing ongoing sub-optimal screening during implementation monitoring.

Third, our findings estimate a low test positivity when compared to 2021 [24]. We speculate that due to non-capture of number of presumptive TB in *Ni-kshay*, percentage presumptive among screened and percentage tested among presumptive cannot be generated. Hence, all the focus in the field in 2021 might have been on test positivity. In the event of not using ACF tool in the field and the absence of mechanisms to cross-check ACF-detected TB (whether they are truly ACF-detected), it is difficult to cross-verify test positivity reported in 2021. In the past, performance of ACF was also being routinely assessed using two methods that focused on ACF-detected TB [30]: i) crude proportion of TB cases detected

by PCF compared to ACF for a defined geographical area and period and ii) additional number of TB cases detected in a geographic area (before-during). These methods of monitoring have many limitations and should be done away with [30].

Overall, low scale makes ACF ineffective and high NNS makes ACF inefficient with already constrained resources. The concern of low yield due to scale and quality being below the benchmark has been repeatedly highlighted in previous assessments of NTEP [31].

By specific high-risk groups, the elderly and prison inmates in our study were an exception where ≥5% screened were identified as presumptive TB. NSS was within or around 1538 cut-off for prisons, contact tracing, non-communicable diseases and people who have recovered from COVID19. A systematic review (2006–19) has reported an NNS of 35 among contacts screened in low- and middle-income countries [6]. Another systematic review from low- and middle-income countries (1999–2020) has revealed that yield of ACF is high in prison, contacts, HIV, drug users and poor households [7]. By broad high-risk groups, geographically high-risk were screened the maximum in our study along with the highest NNS. The high NNS was due to lower percentage of screened identified as presumptive TB, lower test positivity and less use of rapid molecular diagnostic tests.

When compared to routine case-finding within NTEP (most being PCF), the test positivity in ACF care cascade was lower than PCF (1.6% versus 4.5%) [12]. The extent of use of rapid molecular diagnostic tests among ACF- and PCF-detected presumptive TB was similar (26% versus 23%) [12].

## Strengths and limitations

A systematic review assessing the effect of ACF on TB epidemiology recommended that ACF projects should incorporate a well-designed, robust evaluation to contribute to the evidence base [7]. This is one such example. Similar to our 2021 analysis [24], we analysed the ACF scale and quality indicators for an ACF cycle starting from high-risk populations; we did not merely start our analysis from those screened. This provided us with a holistic understanding of low yield, so that corrective actions can be taken. We mapped high-risk populations and prospectively documented ACF care cascade, making it more reliable than previous secondary retrospective aggregate data-based analysis [24,25].

There were three limitations. First, we captured the extent of use of ACF tool by ACF activity days (1031 of 1467 ACF activity days) and not based on what proportion of *Ni-kshay* reported ACF care cascade data were captured by the ACF tool. The reason was that most districts did not update their *Ni-kshay* ACF data for the study period till the completion of our analysis. Second, as we had project research assistants posted in the study districts, findings could present a best-case scenario with the true picture being worse. Finally, this being analysis of routinely captured data, data collection errors cannot be ruled out. The research assistants tried to minimize this when they compiled the forms at district level.

## Policy and program implications

In addition to recommendations from the first and second (qualitative) phases of the project published elsewhere [24,29], the implications from the third phase are as follows.

First, the reasons for ACF scale and quality being below the benchmark were explored in an ongoing systematic qualitative enquiry (second phase of project: February to August 2023) Among NTEP staff at various levels, inadequate training, a significant knowledge gap and know-do gap was observed [29]. Six tips for improving scale and quality of ACF were recommended [29].

Second, more effective planning and resources are required if we are to cover 90% of the high-risk populations at least once a year. There is scope to screen a larger number than the current average of 565 per ACF activity day per district. In resource constrained settings, specific high-risk groups (prison, contacts) may be prioritized.

Third, continuous monitoring of various ACF quality indicators and increased access to rapid molecular diagnostic tests are recommended with a focus on geographically high-risk. To monitor percentage presumptive among screened and percentage tested among presumptive, aggregate number of presumptive TB should be captured against each ACF activity in the ACF module of *Ni-kshay*.

Finally, our findings provide evidence in favour of using an ACF tool for individual level data collection in the field (see S1 File) so that reliable aggregate numbers can be reported in *Ni-kshay*. These can be randomly cross-checked. Additionally, ACF tool may be of use to disburse quality-control linked incentives [29]. Quality-control linked incentives should be for every household visited and every 'sputum collection and transport' (and not linked to number of ACF-detected person with TB) [32,33]. To initiate release of incentives for household visit and sputum collection and transport for a period, as recommended in phase two of the project, completed ACF tool should be submitted, with verification of a certain percentage of data (includes verifying all ACF-detected were detected as a result of ACF) and meeting a conservative test positivity or NNS cut-off [29]. This was successfully done in project *Axshya* (before 2017), where ACF was implemented by non-governmental organizations in India [32]. Decisions about the use of such approaches must be considered in the planning phase [33].

### Broader global relevance

More resources and effective planning, use of ACF tool for individual level data collection in the field, continuous monitoring of various ACF quality indicators, increased access to rapid molecular diagnostic tests and addressing the knowledge/know-do gap are applicable not only for India but also other high TB burden countries that plan to scale up ACF and improve quality in routine program settings. This is applicable irrespective of the ACF algorithm used. These planning and implementation barriers, if unaddressed, will impact the attainment of optimal yield even in the context of adding hand held radiography with computer assisted diagnostics and rapid molecular tests for screening high-risk populations (in addition to symptom screen) [3]. The TB officers at district and state/province level, the national TB programs and their partners (non-Governmental organizations, funding agencies) supporting ACF activities should keep these in mind while funding, planning and implementing ACF.

As correctly stated by MacPherson P et al, while *ACF cannot and should not be a substitute for equitable access to responsive, affordable, accessible primary care services for all*, *with careful planning and substantial investment, ACF for TB can be an impactful approach to accelerating progress towards ending TB in high burden countries* [34].

### Conclusion

In India, using prospective individual level data of TB ACF among high-risk populations in 2023, this study provided nationally representative estimates of all the ACF scale and quality indicators, the time taken and the use of rapid molecular diagnostic tests in the care cascade. This study confirmed the previous findings of one cycle per year and ACF quality being below the benchmark (2021) [24]. Additionally, low scale of ACF was confirmed in our study. The use of rapid molecular tests among ACF-detected presumptive TB was similar to PCF-detected presumptive TB. Individual level data collection of those screened facilitates implementation monitoring of routinely reported aggregate data in *Ni-kshay* ACF module, resulting in reliable indicators. Therefore, the findings of this study are more reliable and supersede the findings of 2021 that were based on retrospective secondary aggregate data [24]. To influence TB transmission, at least one ACF cycle a year should be implemented at scale and with good quality. More effective planning and resources including access to rapid molecular diagnostic tests are recommended.

### Supporting information

**S1 File. ACF tool for individual level data collection in the field.**
(PDF)

**S1 Fig. Thirty randomly sampled NTEP districts from nine states of India\*, phase three of central TB division commissioned national level TB ACF evaluation project, India.** \* Map created using QGIS software using publicly

available and free of cost shape file of districts of India (basefile) from Survey of India website (https://onlinemaps.survey-ofindia.gov.in/).
(DOCX)

**S1 Table. The three revised ACF scale and quality indicators at national level (2021) and their cut-offs used in the first phase of TB ACF evaluation project [17].**
(DOCX)

**S2 Table. Time taken in the care cascade in the TB ACF cycle conducted from January to September 2023 in 30 randomly sampled NTEP districts of India.**
(DOCX)

## Acknowledgments

The India central TB division commissioned, USAID/JSI-supported, ICMR-National Institute of Epidemiology (ICMR-NIE) led, national level TB ACF evaluation project was a collaboration among several key institutions: ICMR-National Institute of Epidemiology (ICMR-NIE) in Chennai, India, USAID India and JSI India, both based in New Delhi; The WHO Country Office for India, also located in New Delhi; and the Central TB Division, Ministry of Health and Family Welfare, Government of India, situated in New Delhi. The authors express sincere gratitude for the invaluable support and contributions from all state TB cells, district TB cells, and the WHO NTEP medical consultant network in India. Additionally, we acknowledge the significant contributions made by research assistants and interns from ICMR-NIE, Chennai, India. The research assistants facilitated data capture in the study districts, while the interns provided essential monitoring of project activities and vital data management support. We acknowledge the following research assistants (not included in the author list): Mr Chandan Kumar Mishra, Mr Deepak Kumar, Ms Santilahun Nongstein, Mr K Arul, Mr Gathala Subhash, Mr Debjyoti Chatterjee, Mr M Rajasekar, Mr Pandiyaraja, Mr Bnajingsuk Nahain Kyndail, Mr Jerry Hujon, Ms Daisy Daimary, Ms Divya Gupta, Ms Krishna Vaghela and Ms Khusbu Ahmas. We acknowledge the following ICMR-NIE interns: Ms. Divya V, Mr. Sathish Kumar, Ms. Suhana Khatoon, Dr. Kavya Rao Bagati, Ms. Lathika R, Ms. Alifia PS, Dr. Aishwarya Dhumale, Dr. Shwetha R, Ms. Rebecca John, Ms. N Bhooma, Ms. B Suvetha, Mr. Mahesh Gomasa, Mr. Divakar G, Mr. Vishnu Bagavath, and Ms VK Damini. We acknowledge the support of KhushiBaby in supporting mobile-based ACF tool for use in the field (part of their Community Health Integrated Platform) in three districts of a state.

## Author contributions

**Conceptualization:** Hemant Deepak Shewade, Amar N Shah, Bhavin Vadera, Venkatesh Roddawar, Sanjay K Mattoo, Swati Iyer, Dheeraj Tumu, Shanmugasundaram Devika, Rakesh R Vaidya, Kiran Rade, Raghuram Rao, Manoj V Murhekar.

**Data curation:** Hemant Deepak Shewade, S Kiran Pradeep, Rakesh R Vaidya, Prasanta Kumar Hota, Asha Frederick, Pankaj Singh, Bayarilin Shanpru, Gunjan Khunger, Subrata Kumar Panda, Mohamed Arif Baig, K V Suma, Mahendran Suleka, Amit Kumar Digal, Debobrata Banerjee, Machupalli Lakshmi Prasanna, Divya Yashpal Waghela, A Krishnaraj, Pratibha Kashyap, Jayeshbhai Sendhabhai Parmar, Sushmita Das, Ajit Kumar, Ajay Kumar, Sunny Yadav, Shradha Chetri, Subhashree Suchismita Mahapatra, Amit Kumar, Monu Pathak, Sachin Singh, Shams Tabrez, Piyush Mehra, Bipra Bishnu, G Mahesh, Biswabihari Mohanty, A Rajesham, Bal Krishna Mishra, Dinesh N Barot, Umesh Chandra Tripathi, Khalid Umer Khayyam.

**Formal analysis:** Hemant Deepak Shewade, S Kiran Pradeep, Prabhadevi Ravichandran.

**Funding acquisition:** Hemant Deepak Shewade, Venkatesh Roddawar.

**Investigation:** Hemant Deepak Shewade, S Kiran Pradeep.

**Methodology:** Hemant Deepak Shewade, S Kiran Pradeep.

**Project administration:** Hemant Deepak Shewade, S Kiran Pradeep, Prabhadevi Ravichandran.

**Resources:** Hemant Deepak Shewade, S Kiran Pradeep.

**Software:** Hemant Deepak Shewade, S Kiran Pradeep, R Sabarinathan.

**Supervision:** Hemant Deepak Shewade, S Kiran Pradeep, Prabhadevi Ravichandran.

**Validation:** Hemant Deepak Shewade, S Kiran Pradeep.

**Visualization:** Hemant Deepak Shewade, S Kiran Pradeep.

**Writing – original draft:** Hemant Deepak Shewade, S Kiran Pradeep, Prabhadevi Ravichandran.

**Writing – review & editing:** Hemant Deepak Shewade, S Kiran Pradeep, G Kiruthika, Amar N Shah, Bhavin Vadera, R Sabarinathan, Venkatesh Roddawar, Sanjay K Mattoo, Swati Iyer, Dheeraj Tumu, Aniket Chowdhury, Shanmugasundaram Devika, Joshua Chadwick, Rakesh R Vaidya, Prasanta Kumar Hota, Asha Frederick, Pankaj Singh, Bayarilin Shanpru, Gunjan Khunger, Subrata Kumar Panda, Mohamed Arif Baig, K V Suma, Mahendran Suleka, Amit Kumar Digal, Debobrata Banerjee, Machupalli Lakshmi Prasanna, Divya Yashpal Waghela, A Krishnaraj, Pratibha Kashyap, Jayeshbhai Sendhabhai Parmar, Sushmita Das, Ajit Kumar, Ajay Kumar, Sunny Yadav, Shradha Chetri, Subhashree Suchismita Mahapatra, Amit Kumar, Monu Pathak, Sachin Singh, Shams Tabrez, Piyush Mehra, Bipra Bishnu, G Mahesh, Biswabihari Mohanty, A Rajesham, Bal Krishna Mishra, Dinesh N Barot, Umesh Chandra Tripathi, Khalid Umer Khayyam, Kiran Rade, Raghuram Rao, Manoj V Murhekar.

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
