## [Decision Letter · Decision Letter 0]

4 Jul 2025

PGPH-D-25-01505

Active case-finding for TB in India: Assessment of scale and quality benchmarks, time taken and use of rapid molecular diagnostic tests

Dear Dr. Shevade,

Thank you for submitting your manuscript to PLOS Global Public Health. After careful consideration, we feel that it has merit but does not fully meet PLOS Global Public Health’s publication criteria as it currently stands. Therefore, we invite you to submit a revised version of the manuscript that addresses the points raised during the review process.

We look forward to receiving your revised manuscript.

Kind regards,

Sachin Atre, Ph.D.

Academic Editor

Additional Editor Comments (if provided):

The manuscript is based on nationally representative individual data on Active Case Finding (ACF) strategy and its implementation. It is important. But it showed that despite massive effort, the ACF implementation is not detecting any significant number of TB cases. On the contrary, observations in their both cycles showed that the Passive Case Finding (PCF) is superior over the ACF. Considering this evidence and limited resources available for the NTEP, ideally ACF should be discouraged by the NTEP. Authors are advocating for the ACF and making some generic recommendations like monitoring, additional resource investments, proper implementation etc which is somehow paradoxical. The evidence clearly suggests that the NTEP should continue with passive case finding which is a more feasible and cost effective strategy. Authors may plan to undertake a major revision especially for discussion and conclusion sections in the light of the above observations. This is a fundamental point for revision.

Other comments are as below

1. Line 51: Specify the name of the program: NTEP

2. Line 52: Algorithm of what?

3. Lines 196-197: Please clarify the unit for these numbers

4. Lines 205-206: "During the national/state/district level orientation and re-orientation meetings and in writing,

it was clarified that individual level data available from the ACF tool used in field (see

Supplementary Annex) will be considered as ACF data. "Who claimed?

5. ACF diagnosed 1.6 % of cases and Passive case Finding (PCF) diagnosed 4.5% cases and use of rapid molecular tests were almost similar. It showed that PCF is superior than the ACF due to high yield and low resource requirement.

6. The suggestions are generic and not based on findings of the study.

7. The manuscript requires significant editing to make it publishable.

Reviewers' comments:

Reviewer's Responses to Questions

**Comments to the Author**

1. Does this manuscript meet PLOS Global Public Health’s publication criteria ? Is the manuscript technically sound, and do the data support the conclusions? The manuscript must describe methodologically and ethically rigorous research with conclusions that are appropriately drawn based on the data presented.

Reviewer #1: Yes

Reviewer #2: Yes

2. Has the statistical analysis been performed appropriately and rigorously?

Reviewer #1: Yes

Reviewer #2: Yes

3. Have the authors made all data underlying the findings in their manuscript fully available (please refer to the Data Availability Statement at the start of the manuscript PDF file)?

Reviewer #1: Yes

Reviewer #2: Yes

4. Is the manuscript presented in an intelligible fashion and written in standard English?

Reviewer #1: Yes

Reviewer #2: Yes

5. Review Comments to the Author

Reviewer #1: Thank you for sharing the manuscript based on analysis of ACF data. It highlights an important aspect of significant strategy for case finding through analyzing secondary data. The manuscript is well drafted and covers important issues in implementing ACF strategies. However, there are few points to consider to further improve the manuscript.

1. Introduction section

Strengths:

• This study fills a vital evidence gap by providing nationally representative, prospectively collected, individual-level data on TB active case finding (ACF) a major upgrade from earlier reliance on local or retrospective aggregate data.

• The focus on scale and quality benchmarks and inclusion of rapid molecular diagnostic test usage makes it timely and highly relevant in the post-COVID TB control landscape.

Suggestion for improvement:

• While the novelty is high, the research question is largely evaluative and programmatic rather than conceptual or hypothesis-driven. There is a scope for adding relevant modeling to make it theoretically robust.

• Introduction section need to be more tightened, the references need to be carefully sited

• Make the objectives more self-explanatory.

2. Methodology

Strengths:

• The design—a prospective cohort study using standardized tools (paper/mobile-based ACF tool)—ensures a high degree of data fidelity.

• The sampling strategy is robust, covering 30 districts across nine Indian states, stratified by performance tiers and selected by probability proportionate to size.

Suggestions:

• There is limited explanation of how quality assurance was maintained across the data collection chain—especially given the scale and reliance on district-level staff.

• ACF coverage was calculated using an assumption (75% of mapped population during Jan–Sept)—this operational workaround may skew the interpretation of scale.

• Though high-risk group has been explained in table but it needs to be explained in the text as well.

• Suggesting to add details about quality assurance.

3. Results and Interpretation

Strengths:

• The results are detailed, well-tabulated, and stratified across multiple high-risk groups—providing granular insights.

• The number needed to screen (NNS) calculation is a practical composite measure summarizing ACF efficiency.

Suggestions:

• The paper rightly critiques the low performance but stops short of offering quantitative modeling of the potential impact of scale-up or quality improvement on TB notifications or prevalence.

• Adding more information about qualitative indicators can make it more robust if available.

4. Discussion and Policy Implications

Suggestions:

• Some conclusions repeat content from the results and could be more concise by adding more concluding statements.

• Conclusion need to be re-written making it more conclusive and interpretative in terms of presenting the results.

Overall comments

The author list is extensive (possibly over 60 authors)—which raises questions about authorship criteria, especially given the emphasis on field staff in acknowledgments.

• The abstract has become quite descriptive and could be shortened by removing some redundant statistics to enhance clarity.

Recommendations for Improvement

1. Add statistical modeling or sensitivity analyses to project the potential impact of meeting benchmark targets (e.g., if ACF achieved 90% coverage).

2. Include more discussion on implementation barriers, especially in low-resource districts—could be based on data from the second qualitative phase.

3. Reframe implications for global health programs—tie in learnings from other high-burden countries or WHO strategies.

4. Clarify authorship rationale, particularly the roles of field-level personnel who contributed significantly but are listed outside the author list.

Reviewer #2: This is a well-timed study that provides detailed assessment of the ACF approach—highlighting specific areas for improvement—and proposes actionable, evidence-based suggestions to enhance its effectiveness.

Overall comment: You may want to include how this recommendations can be taken forward within the TB programme. what strategy do you envision for integrating this recommendation into TB programme planning and execution?)

Please refer to the specific in-text comments in the manuscript for details.

6. PLOS authors have the option to publish the peer review history of their article (what does this mean? ). If published, this will include your full peer review and any attached files.

**Do you want your identity to be public for this peer review?** For information about this choice, including consent withdrawal, please see our Privacy Policy .

Reviewer #1: **Yes: ** Prashant Subhash Kulkarni

Reviewer #2: No

---

## [Decision Letter · Decision Letter 1]

24 Sep 2025

PGPH-D-25-01505R1

Active case-finding for TB in India: Assessment of scale and quality benchmarks, time taken and use of rapid molecular diagnostic tests

Dear Dr. Shewade

Thank you for submitting your manuscript to PLOS Global Public Health. After careful consideration, we feel that it has merit but does not fully meet PLOS Global Public Health’s publication criteria as it currently stands. Therefore, we invite you to submit a revised version of the manuscript that addresses the points raised during the review process.

We look forward to receiving your revised manuscript.

Kind regards,

Sachin Atre, Ph.D.

Academic Editor

Journal Requirements:

Additional Editor Comments (if provided):

Editor: All comments are addressed.

Reviewer #1: Revise "Introduction section" especially the first paragraph which has a few unclear statements. The discussion section needs to be more structured.

Reviewer #2: Thank you for all the revisions. All my previous comments have been addressed well. I only have a small suggestion regarding the first sentence of the abstract.

Line no: 50-51-Since 2017, tuberculosis active case-finding (TB ACF) in India is being implemented in routine national TB elimination program settings. You may consider revising this sentence to make it easier to read- Since 2017, tuberculosis active case-finding (TB ACF) has been implemented within the routine framework of India’s national TB elimination program.

Reviewers' comments:

Reviewer's Responses to Questions

**Comments to the Author**

1. If the authors have adequately addressed your comments raised in a previous round of review and you feel that this manuscript is now acceptable for publication, you may indicate that here to bypass the “Comments to the Author” section, enter your conflict of interest statement in the “Confidential to Editor” section, and submit your "Accept" recommendation.

Reviewer #1: All comments have been addressed

Reviewer #2: All comments have been addressed

2. Does this manuscript meet PLOS Global Public Health’s publication criteria ? Is the manuscript technically sound, and do the data support the conclusions? The manuscript must describe methodologically and ethically rigorous research with conclusions that are appropriately drawn based on the data presented.

Reviewer #1: Yes

Reviewer #2: Yes

3. Has the statistical analysis been performed appropriately and rigorously?

Reviewer #1: Yes

Reviewer #2: Yes

4. Have the authors made all data underlying the findings in their manuscript fully available (please refer to the Data Availability Statement at the start of the manuscript PDF file)?

Reviewer #1: Yes

Reviewer #2: Yes

5. Is the manuscript presented in an intelligible fashion and written in standard English?

Reviewer #1: Yes

Reviewer #2: Yes

6. Review Comments to the Author

Reviewer #1: Revise Introduction section especially the first paragraph which has few unclear statements. The discussion section needs to be more structured.

Reviewer #2: Thank you for all the revisions. All my previous comments have been addressed well. I only have a small suggestion regarding the first sentence of the abstract; otherwise, the manuscript looks ready.

Line no: 50-51-Since 2017, tuberculosis active case-finding (TB ACF) in India is being implemented in routine national TB elimination program settings.

You may consider revising this sentence to make it easier to read- Since 2017, tuberculosis active case-finding (TB ACF) has been implemented within the routine framework of India’s national TB elimination program.

7. PLOS authors have the option to publish the peer review history of their article (what does this mean? ). If published, this will include your full peer review and any attached files.

**Do you want your identity to be public for this peer review?** For information about this choice, including consent withdrawal, please see our Privacy Policy .

Reviewer #1: **Yes: ** Prashant Kulkarni

Reviewer #2: No

 Figure Resubmissions:

---

## [Decision Letter · Decision Letter 2]

7 Oct 2025

Active case-finding for TB in India: Assessment of scale and quality benchmarks, time taken and use of rapid molecular diagnostic tests

PGPH-D-25-01505R2

Dear Dr. Shewade

We are pleased to inform you that your manuscript 'Active case-finding for TB in India: Assessment of scale and quality benchmarks, time taken and use of rapid molecular diagnostic tests' has been provisionally accepted for publication in PLOS Global Public Health.

Best regards,

Sachin Atre, Ph.D.

Academic Editor

Reviewer 1: This paper offers critical evidence for mid-course correction in India’s TB elimination program. Visual simplification of figures/tables is suggested.

Reviewer Comments (if any, and for reference):

Reviewer's Responses to Questions

**Comments to the Author**

1. If the authors have adequately addressed your comments raised in a previous round of review and you feel that this manuscript is now acceptable for publication, you may indicate that here to bypass the “Comments to the Author” section, enter your conflict of interest statement in the “Confidential to Editor” section, and submit your "Accept" recommendation.

Reviewer #1: All comments have been addressed

2. Does this manuscript meet PLOS Global Public Health’s publication criteria ? Is the manuscript technically sound, and do the data support the conclusions? The manuscript must describe methodologically and ethically rigorous research with conclusions that are appropriately drawn based on the data presented.

Reviewer #1: Yes

3. Has the statistical analysis been performed appropriately and rigorously?

Reviewer #1: Yes

4. Have the authors made all data underlying the findings in their manuscript fully available (please refer to the Data Availability Statement at the start of the manuscript PDF file)?

Reviewer #1: Yes

5. Is the manuscript presented in an intelligible fashion and written in standard English?

Reviewer #1: Yes

6. Review Comments to the Author

Reviewer #1: This paper offers critical evidence for mid-course correction in India’s TB elimination program. Visual simplification of figures/tables is suggested.

7. PLOS authors have the option to publish the peer review history of their article (what does this mean? ). If published, this will include your full peer review and any attached files.

**Do you want your identity to be public for this peer review?** For information about this choice, including consent withdrawal, please see our Privacy Policy .

Reviewer #1: **Yes: ** Prashant Subhash Kulkarni
